# Chimeric Antigen Receptor T-Cell Therapy: The Light of Day for Osteosarcoma

**DOI:** 10.3390/cancers13174469

**Published:** 2021-09-05

**Authors:** Zili Lin, Ziyi Wu, Wei Luo

**Affiliations:** Department of Orthopaedics, Xiangya Hospital, Central South University, Changsha 410008, China; woslinzili@163.com (Z.L.); wzy15021@163.com (Z.W.)

**Keywords:** osteosarcoma, CAR-T therapy, solid tumors, immune checkpoint, targeted therapy

## Abstract

**Simple Summary:**

As a novel immunotherapy, chimeric antigen receptor (CAR) T-cell therapy has achieved encouraging results in leukemia and lymphoma. Furthermore, CAR-T cells have been explored in the treatment of osteosarcoma (OS). However, there is no strong comprehensive evidence to support their efficacy. Therefore, we reviewed the current evidence on CAR-T cells for OS to demonstrate their feasibility and provide new options for the treatment of OS.

**Abstract:**

Osteosarcoma (OS) is the most common malignant bone tumor, arising mainly in children and adolescents. With the introduction of multiagent chemotherapy, the treatments of OS have remarkably improved, but the prognosis for patients with metastases is still poor, with a five-year survival rate of 20%. In addition, adverse effects brought by traditional treatments, including radical surgery and systemic chemotherapy, may seriously affect the survival quality of patients. Therefore, new treatments for OS await exploitation. As a novel immunotherapy, chimeric antigen receptor (CAR) T-cell therapy has achieved encouraging results in treating cancer in recent years, especially in leukemia and lymphoma. Furthermore, researchers have recently focused on CAR-T therapy in solid tumors, including OS. In this review, we summarize the safety, specificity, and clinical transformation of the targets in treating OS and point out the direction for further research.

## 1. Introduction

Osteosarcoma (OS) is the most observed malignant primary bone tumor and chiefly affects children and adolescents [1]. With the introduction of multiagent chemotherapy, the long-term overall survival rate for OS varies from 60–70% [2]. However, the prognosis for patients with metastatic or relapsed OS has remained bleak and stagnant over the past 30 years, with a five-year overall survival rate of 20% [3,4,5]. In addition, the adverse effects brought by conventional treatments, including radical surgery and systemic chemotherapy, seriously affect the survival quality of patients. Therefore, new treatments for OS are urgently needed. Over the past few decades, immunotherapy has played an increasingly important role in treating cancer [6], with monoclonal antibodies (MABs) and chimeric antigen receptors (CARs) receiving much attention [7,8].

CARs are engineered receptors composed by an extracellular single-chain variable fragment (scFv) derived from a monoclonal antibody, a transmembrane domain, and an intracellular domain derived from the T-cell receptor CD3-ζ chain. In the lab, technicians activate T cells and genetically engineer CARs onto their membranes, transforming T cells into chimeric antigen receptor T (CAR-T) cells (Figure 1). The intracellular domains of CAR-T cells can bind to costimulatory molecules, such as CD28 and 41BB. The CARs without costimulatory molecules are the first-generation CARs, the CARs with one costimulatory molecule are the second-generation CARs, and the CARs with two costimulatory molecules are the third-generation CARs [9]. CARs transfer specific antigens to immune effector cells (typically T cells), providing them with the ability to target tumors [10]. Therefore, CARs can enable T cells to produce potent antitumor activity without the limitation of traditional major histocompatibility complexes (MHCs) [11,12,13]. Compared with traditional therapies, CARs have the following advantages: (1) cells loaded with CARs can target tumor tissues, leading to more precise and less invasive therapy; (2) cells loaded with CARs can still have a strong lethality to metastatic or recurrent tumor cells [14]. Compared with MABs, CARs have the following advantages: (1) MABs can fail to recognize tumor cells when they express too little antigen, while CARs can overcome this dilemma; (2) immune cells loaded with CARs can not only kill tumor cells directly but also secrete cytokines, such as IFN-γ and IL-2, exerting powerful antitumor effects; (3) immune cells loaded with CARs can proliferate massively and exert long-lasting anti-tumor effects after infusion [13,15,16,17]. Taken together, CARs may offer a promising approach for the treatments of cancer.

Over the years, CAR-T cells have gained prominence in treating cancers, especially leukemia and lymphoma [18]. Significant clinical responses and high complete response rates have been observed in CAR-T therapy for B-cell malignancies. Based on these encouraging results, the Food and Drug Administration (FDA) recently approved two CAR-T cells targeting the CD19 protein for the treatment of acute lymphoblastic leukemia and diffuse large B-cell lymphoma [19]. Recently, researchers have also extended CAR-T therapy to solid tumors, including OS [20,21,22]. Koksal et al. summarized CAR-T-cell therapy for OS several years ago [22]. However, given that new CAR-T therapies for OS have been emerging in recent years, we feel it necessary to provide a more comprehensive review of CAR-T-cell therapy for OS.

## 2. Research on CAR-T Cells Targeting OS Antigens

The key step in the generation of CAR-T cells is finding and targeting the tumor precisely, reducing damage to normal tissue. Currently, the main targets of CAR-T cells in OS treatment are as follows: human epidermal growth factor receptor 2 (HER2), disialoganglioside (GD2), B7-H3 (clusters of differentiation protein 276, CD276), Interleukin-11 receptor a-chain (IL-11Ra), type I insulin-like growth factor receptor (IGF1R), receptor tyrosine kinase-like orphan receptor 1 (ROR1), erythropoietin-producing hepatocellular receptor tyrosine kinase class A2 (EphA2), natural killer group 2D (NKG2D), activated leukocyte cell adhesion molecule (ALCAM, CD166), and chondroitin sulfate proteoglycan 4 (CSPG4) (Figure 2, Table 1) [23,24,25,26,27,28,29,30].

### 2.1. Human Epidermal Growth Factor Receptor 2 (HER2)

The human epidermal growth factor receptor (HER) family consists of four members, HER1–4, which comprise essentially a class of tyrosine-protein kinases that functions as both homodimer and heterodimer. Their ligands include epidermal growth factor (EGF) and transforming growth factor-α (TGF-α). After binding with ligands, they activate the downstream signaling pathway and produce a series of physiological or pathological reactions. Unlike other members, HER2 does not bind to any ligand and performs its function by forming a heterodimer with other members [31]. The formation of a heterodimer is typically due to HER2 overexpression, leading to autophosphorylation of tyrosine residues in the cytoplasmic domain of the heterodimer and initiating various signaling pathways contributing to cell proliferation and tumorigenesis [56]. HER2-targeted therapy for cancer has been widely studied and has yielded encouraging results in the treatment of breast cancer [32]. In addition, HER2 overexpression has been reported in other solid tumors, such as pancreatic adenocarcinomas, colorectal carcinomas, and gastric cancer [33]. In addition, relevant studies have shown that HER2 is also expressed in OS [57,58,59]; therefore, HER2-targeted therapy in treating OS is worthy of study. Ahmed et al. showed that HER2-targeted CAR-T cells secrete immunostimulatory cytokines and proliferate after exposure to HER2-positive OS cells. The transfer of HER2-targeted CAR-T cells into mouse models with xenografted OS resulted in tumor regression and remission in mouse models with OS lung metastasis [15]. Experiments conducted by Rainusso et al. showed that HER2-targeted CAR-T cells significantly reduced sarcosphere formation capacity and bone tumor growth in immunodeficient mice after orthotopic transplantation. Additionally, they found this in vivo, giving HER2-targeted CAR-T cells markedly reduced drug-resistant tumor-initiating cells in bulky OS [34]. Similarly, HER2-targeted CAR-T cells displayed significant antitumor activity in the study conducted by Park and colleagues [23]. In a phase I/II clinical study, Ahmed et al. evaluated the feasibility and safety of escalating doses of HER2-targeted CAR-T cells in patients with recurrent or refractory sarcomas, including OS. They found that infusion of 1 × 10^8^/m^2^ of HER2-targeted CAR-T cells was well tolerated, with stable disease lasting from 12 weeks to 14 months in 4 of 17 evaluable patients. After removal of residual metastases, three patients remained in remission at 6, 12, and 16 months. Their experiments showed that a safe dose of HER2-targeted CAR-T cells can be established for cancer patients. In addition, HER2-targeted CAR-T cells can be transported to the tumor site and maintained at low levels in a dose-dependent manner for more than 6 weeks [60]. These experiments have confirmed that HER2-targeted CAR-T cells can not only kill tumor-initiating cells in OS that lead to OS recurrence but can also eliminate OS metastatic lesions, which are responsible for the deaths of OS patients. Thus, HER2-targeted CAR-T cells are a potentially worthy treatment. However, although Ahmed et al. demonstrated the safety of HER2-targeted CAR-T cells, Morgan RA’s case report is of concern. In the Morgan RA case report, a patient died after a cytokine storm following treatment with HER2-targeted CAR-T cells [61]. To address this problem, Mata et al. created a canine HER2-targeted CAR-T cell. They found that although the cytolytic activity of canine HER2-targeted CAR-T cells was similar to their previous studies using human HER2-targeted CAR-T cells [15], cytokine production was lower [62]. This large animal model may be valuable in evaluating the development of CAR-T cells for future human clinical trials. Therefore, to transform HER2-targeted CAR-T cells into clinical usage, strategies to address their adverse effects should be investigated. In view of the cytokine storm, we can introduce a coexpressed suicide gene or inhibitive receptors into CAR-T cells to properly limit their functions.

### 2.2. Disialoganglioside (GD2)

Disialoganglioside (GD2), an N-acetyl neuraminic acid-containing glycolipid antigen, is composed of five monosaccharides anchored to the lipid bilayer of the plasma membrane through a ceramide lipid [35]. GD2 expression is low in normal tissues [36] but high in solid tumors in children and adults, including neuroblastomas, Ewing’s sarcoma, and OS [63]. GD2 is often used in treating neuroblastomas and has achieved encouraging results [64]. The study by Roth et al. showed that OS can express more GD2 than neuroblastomas [65]. Therefore, targeting GD2 for OS seems reasonable. Long et al. observed 100% expression of GD2 in OS and that third-generation GD2-targeted CAR-T cells mediated efficient and comparable lysis of GD2^+^ sarcoma cell lines in vitro [37]. In another experiment, Chulanetra et al. demonstrated that GD2-targeted CAR-T cells were very effective in inducing OS tumor cell death. In further experiments, they showed that suboptimal treatment with doxorubicin can effectively kill OS tumor cells in conjunction with CAR-T cells [38]. Park and Cheung also demonstrated that GD2-targeted CAR-T-cell therapy enabled effective T cells to infiltrate tumors and exert potent antitumor activity in OS. Moreover, there was more T-cell infiltration in tumors when combined with anti-PD-L1 (programmed death ligand 1), especially when anti-PD-L1 was applied after GD2-targeted therapy [23]. However, although GD2-targeted CAR-T-cell therapy is a promising approach for treating OS, there are still many problems that still need to be solved. Long et al. found that GD2-targeted CAR-T cells had no antitumor effect against GD2^+^ sarcoma in xenograft models. They further found that xenografts from pediatric sarcoma induced myeloid-derived suppressor cells (MDSCs) to inhibit human CAR-T-cell responses in vitro. However, this inhibition disappeared with the addition of all-trans retinoic acid, suggesting that all-trans retinoic acid improved the antitumor ability of GD2-targeted CAR-T cells [37]. In addition, Chulanetra et al. discovered that after interacting with GD2-targeted CAR-T cells, OS cells can upregulate PD-L1 expression. Moreover, this specific interaction induced CAR-T cells to overexpress the exhaustion marker PD-1, leading to CAR-T-cell apoptosis [38], supporting the experimental conclusions of Park and Cheung [23]. Therefore, improving the efficiency of GD2-targeted CAR-T cells is essential to its transformation into the clinic. One strategy is to combine GD2-targeted CAR-T cells with immune checkpoint blockade. Related studies have shown that combining GD2-targeted CAR-T cells and anti-PD-L1 can improve the antitumor effect of GD2-targeted CAR-T cells [23]. Furthermore, the ability of other immune checkpoint blockades to coordinate with GD2-targeted CAR-T cells is also worthy of exploring. Another strategy is to construct bispecific CAR-T cells. Various studies have shown that MDSCs can inhibit CAR-T cell responses and express the NKG2D ligand [66]. Thus, constructing GD2-targeted CAR-T cells loaded with NKG2D may be a viable method to strengthen the function of GD2-targeted CAR-T cells. In addition, Kailayangiri et al. reported that inhibition of enhancer of zeste homolog 2 (EZH2) can upregulate tumor antigen GD2 synthase expression to upregulate the expression of GD2 [67]. Therefore, upregulating the tumor antigen expressed in OS by epigenetics is a viable approach.

### 2.3. B7-H3 (CD276, Clusters of Differentiation Protein 276)

B7-H3 is an immune checkpoint in the B7 family of molecules that interacts with checkpoint markers, such as CTLA4 (cytotoxic T lymphocyte-associated antigen 4) and PD-1. Human B7-H3 protein exists in transmembrane or soluble form. Transmembrane B7-H3 is present not only on the surface of tumor cells but also in cytoplasmic vesicles and the nucleus [39]. Related studies have suggested that B7-H3 plays an essential role in the proliferation, invasion, and migration of malignant tumor cells [68,69,70,71]; therefore, B7-H3 can be used as a target for tumor immunotherapy. Indeed, B7-H3 has been used in many cancers, including acute myeloid leukemia, non-small-cell lung cancer, and neuroblastoma [72,73,74]. Moreover, B7-H3 is expressed in OS, and 8H9, a MAB targeting B7-H3, has also been used against OS [75,76,77,78]. Therefore, research on treating OS with B7-H3-targeted CAR-T is also being explored. Majzner et al. showed that systemic administration of B7-H3-targeted CAR-T cells mediated the regression and eradication of OS xenografts. In addition, a model of the highly metastatic OS with a 100% lethality survived almost completely after treatment with B7-H3-targeted CAR-T cells [29]. Murty et al. showed that B7H3-targeted CAR-T cells had an antitumor effect on naturally expressing B7H3 OS cells in an in vitro co-culture assay. Furthermore, an advanced orthotopic model of human OS treated with B7-H3-targeted CAR-T cells showed significantly reduced tumor volume with prolonged overall survival [40]. Therefore, laboratory evidence has suggested that B7-H3-targeted CAR-T cells have the potential as a therapeutic approach for OS and exploring methods to improve this effect can speed the clinical transformation. Related studies have reported that B7-H3 may participate in the JAK/STAT and PI3K/Akt/mTOR pathways and that the inhibition of B7-H3 increases the response of tumor cells to inhibitors of these pathways [79,80]. Consequently, combining B7-H3-targeted CAR-T cells with chemotherapy may serve as a synergistic treatment approach.

### 2.4. Interleukin-11 Receptor a-Chain (IL-11Ra)

IL-11, a member of the family of pleiotropic cytokines, is both a pro- and an anti-inflammatory cytokine [81,82]. The specific binding of IL-11 to IL-11Ra mediates the assembly of a multi-subunit receptor complex. Then, the receptor complex can initiate intracellular signaling through association with the transmembrane signal transducer glycoprotein gp-130, leading to a series of physiological and pathological responses [82,83]. IL-11Ra is involved in the development and aggressive activity of various cancers [41,84]. In recent years, the role of IL-11Ra in OS has also attracted attention [85]. Lewis et al. found that IL-11Rα was significantly expressed in primary OS and pulmonary metastatic OS in an orthotopic mouse model but absent from the control normal tibia and lung. Moreover, IL-11Rα expression in large samples of human primary and metastatic OS was significantly consistent with the observed results in an orthotopic mouse model [86]. Similarly, Liu et al. reported that OS can highly express IL-11Rα and that near-infrared labeled IL-11Rα imaging agents can detect OS in mouse tumor xenografts [87]. In one experiment, G. Huang and colleagues demonstrated that OS cell lines and OS lung metastases express IL-11Ra. Furthermore, they engineered IL-11Ra-targeted CAR-T cells, demonstrating that these cells kill OS cells and accumulated in lung metastases rather than in surrounding normal lung tissue. However, because IL-11Ra is not 100% expressed on each osteosarcoma cell, it needs to be combined with other therapies to improve its efficacy [24]. In addition, constructing bispecific CAR-T cells is also an alternative method.

### 2.5. Type I Insulin-Like Growth Factor Receptor (IGF1R)

IGF1R is a transmembrane glycoprotein that stimulates the growth of tumor cells by autocrine signaling, induces metastasis, and inhibits apoptosis [42,43,88]. IGF1R overexpression has been found in many human cancers, including OS, and the application of IGF1R in treating OS has also been explored [88,89,90]. In one study, researchers found that OS cells, including those resistant to various conventional anticancer drugs, are sensitive to IGF1R inhibitors. Blocking IGF1R could inhibit proliferation and induce apoptosis of OS cell lines [91]. A recent study evaluated IGF1R-targeted CAR-T cells in sarcomas and found that IGF1R-targeted CAR-T cells remarkedly reduced tumor growth in pre-established, localized, and systemically disseminated OS mouse models. In addition, the adoptive transfer of IGF1R-targeted CAR-T cells also displayed a prolonged survival benefit in a localized sarcoma model [25]. IGF1R is also expressed in normal tissues [43]; therefore, the potential toxicity of IGF1R-targeted therapy should not be ignored. In addressing this problem, we can adjust the affinity between CAR-T cells and the tumor-associated antigens to properly limit the function of CAR-T cells to reduce off-target toxicity.

### 2.6. Receptor Tyrosine Kinase-Like Orphan Receptor 1 (ROR1)

ROR1 is a type I transmembrane protein that is physiologically expressed during early embryogenesis and plays a key role in organogenesis. It is rarely expressed in tissues after birth but is expressed in various cancers, especially those with low differentiation [44]. In recent years, ROR1 has attracted attention as an immune target. Cirmtuzumab, a humanized MAB that blocks ROR1 signaling, was effective and safe in patients with advanced, relapsed, or refractory chronic lymphoblastic leukemia in a phase I trial [44]. One study revealed that ROR1 MAB markedly blocked the migration of OS cells, suggesting ROR1 may be a novel therapeutic target to delay OS metastasis [92]. Furthermore, X. Huang et al. demonstrated that ROR1 is highly expressed in sarcoma cell lines, including Ewing’s sarcoma, OS, rhabdomyosarcoma, and fibrosarcoma. In addition, their studies indicated that ROR1-targeted CAR-T cells showed specific cytotoxicity and released mainly IFN-γ, TNF-α, and IL-13 cytokines against sarcomas in vitro. They further found that ROR1-targeted CAR-T cells remarkedly suppressed sarcoma growth in pre-established localized and disseminated sarcoma xenograft models associated with prolonged survival [25]. Although the above experiments proved that ROR1-targeted CAR-T cells may be a promising target for treating OS, the safety of ROR1-targeted CAR-T cells cannot be ignored. Recent studies have shown that ROR1 expression is not specific to tumor tissue. ROR1 has been observed in other normal tissues in humans, high in the gastric antrum and body, although experiments in the macaque model have shown no significant adverse effects [45]. Additionally, the logic-gated ROR1 CAR designed by Srivastava et al. can protect healthy tissues and target tumor cells, promising to address the off-target toxicity [93].

### 2.7. Natural Killer Group 2D (NKG2D)

NKG2D is a powerful activating receptor expressed by natural killer (NK) cells and T cells that are involved in immune responses during infection, cancer, and autoimmunity [46]. In recent years, the role of NKG2D and its ligands in cancer has attracted increasing attention [94,95,96]. MHC class I chain-related molecule A (MICA) is a major ligand for activating immune receptor NKG2D [46,97]. MICA is commonly expressed at mRNA and protein levels in OS. Furthermore, MICA expression was upregulated in OS compared with benign tumors and normal bone tissue. Restoration of NKG2D receptor expression on immune effector cells may contribute to therapeutic strategies for human OS [97,98]. Chang et al. significantly increased the surface expression of NKG2D in NK cells by retroviral transduction of NKG2D-DAP10-CD3z. They found that in immunodeficient mice transplanted with OS cells, NK cells expressing the NKG2D-DAP10-CD3z receptor had great antitumor activity and produced significant tumor reduction, while the simulated transduction-activated NK cells were ineffective [99]. Fernandez et al. extended NKG2D to T cells by creating NKG2D CAR-redirected memory T cells that strengthened cytotoxicity against OS cells in vitro, compared with untransduced T cells. Moreover, in vivo, tumor growth was restricted, and survival was prolonged in mice that received NKG2D CAR-redirected memory T cells. In addition, NKG2D CAR-redirected memory T cells showed no lytic activity against healthy cells and no chromosomal aberrations due to lentiviral transduction [47]. Based on these studies, intervening NKG2D receptors on immune cells may be a strategy for treating OS. However, Van Seggelen et al. recently described lethal toxicity in mice treated with murine NKG2D-redirected CAR-T cells [26], raising safety concerns. Research conducted by Fernandez et al. showed that adult health tissues are not sensitive to NKG2D-redirected CAR-T-cell cytotoxicity [47]. In addition, an ongoing phase I dose-escalation study to test the safety of NKG2D-redirected CAR-T cells in patients with myeloid malignancies did not show any significant adverse effects [100]. Although safety problems have been reported in animal experiments, adverse reactions in human beings have not been reported. However, the safety of NKG2D-redirected CAR-T cells cannot be ignored. We can adjust the affinity between CAR-T cells and the antigen epitopes to suitably limit the function of CAR-T cells to reduce off-target toxicity.

### 2.8. Erythropoietin-Producing Hepatocellular Receptor Tyrosine Kinase Class A2 (EphA2)

EphA2 is a tyrosine kinase receptor involved in ephrin signaling during embryonic development, and its post-developmental expression is mainly confined to some epithelial cells [48,49,50]. EphA2 overexpression has been widely reported in various cancer [101,102] and plays an important role in OS [103]. There are also many drugs targeting EphA2 for treating OS, and they have achieved some efficacy [104,105,106]. Related studies also have suggested the EphA2 receptor may be an attractive candidate receptor for the targeted delivery of therapeutics to OS [107]. Hsu and colleagues showed that EphA2-targeted CAR-T cells effectively killed EphA2-expressing OS tumor cells in pre-established, targeted xenografts in immunodeficient mice and were associated with prolonged survival. In addition, EphA2-targeted CAR-T cells remarkedly reduced or eliminated tumor burden in a mouse model of disseminated OS metastasis. However, EphA2 has its limits, including an increase in EphA2-negative tumor cells after treatment with EphA2-targeted CAR-T cells, which may lead to tumor immune escape [30]. Therefore, EphA2 should be an ideal target for treating OS, but immune evasion after treatment must be addressed. One approach is to construct bispecific CAR-T cells to target EphA2-negative tumor cells. Another approach is to combine EphA2-targeted CAR-T cells with immune checkpoint blockade or chemotherapy.

### 2.9. Activated Leukocyte Cell Adhesion Molecule (ALCAM, CD166)

CD166 (ALCAM), a type I membrane protein, is a member of the immunoglobulin gene superfamily and a ligand for the lymphocyte antigen CD6 [51]. CD166 has three different subtypes, namely membrane CD166, cytoplasmic CD166, and soluble CD166 [108]. Many studies have shown that CD166 is associated with the occurrence and development of various malignancies, including intestinal carcinoma, glioblastoma, and prostate cancer [52,109,110,111]. Previous studies also showed that OS cells can express CD166 [112,113], and targeting OS through CD166 has also achieved a certain effectiveness [114]. Wang et al. showed that OS cell lines express CD166 in varying levels. They further found that CD166-targeted CAR-T cells incorporated with 4-1BB showed cytotoxicity to OS in vitro and in vivo; the cytotoxicity of which was closely related to the expression of CD166 [27]. However, CD166-targeted CAR-T-cell therapy also has some limits. CD166 is present not only in tumors but also in some normal tissues, such as epithelial cells, fibroblasts, and neurons [115,116,117]. Thus, although Wang and his team showed that CD166-targeted CAR-T cells were safe, we still cannot ignore this issue. The aforementioned methods, including adjusting the affinity between CAR-T cells and antigen epitopes or constructing logic-gated CAR, may serve as a solution to this issue.

### 2.10. Chondroitin Sulfate Proteoglycan 4 (CSPG4)

CSPG4, a transmembrane proteoglycan, is lowly expressed in normal tissues but highly expressed in several solid tumors and plays a central tumorigenic role [53,54,55]. In recent years, CSPG4 has become a CAR-target antigen for many different cancer entities [28]. Relevant experiments have shown that both human and canine OS cells highly express CSPG4, and high CSPG4 expression is associated with short survival. CSPG4 immune-targeted therapy for OS can significantly inhibit the proliferation, migration, and osteogenesis of CSPG4-positive OS cells [53,118]. Beard et al. found that CSPG4-targeted CAR-T cells showed cytokine secretion and cytolytic function after co-culture with OS cells [53]. However, evidence for CSPG4-targeted CAR-T cells in treating OS is far from enough; thus, we are not able to accurately evaluate their efficacy. Therefore, more experiments should be conducted.

## 3. Discussion

OS is a primary malignant bone tumor with a global incidence of 3.4 per 1 million people each year [119,120]. Despite multiple changes in adjuvant chemotherapy regimens for patients with OS, the overall survival rate for patients with OS has staggered at 60–70% over the past decades [2]. The prognosis for patients with metastatic or relapsed OS remains poor, with a five-year overall survival rate of 20% [3,4,5]. In addition, the poor targeting of traditional treatment often brings certain adverse effects [121], and the tumor cells develop resistance to chemotherapeutic drugs [122,123]. Therefore, new and effective treatments await exploitation.

CAR-T-cell therapy is a promising immunotherapy approach with encouraging results in treating leukemia and lymphoma [18]. Because of their high targeting and powerful antitumor activity, CAR-T cells are widely used in research and the treatment of various cancers. Recently, researchers have also extended CAR-T therapy to solid tumors, including OS [20,21,22]. Furthermore, related studies have reported that adoptively transferred effector cells derived from naive T cells mediate superior antitumor effects, and youth carry more naive cells [124,125]. Therefore, CAR-T cells may exert more positive effects in treating OS mainly in youth. Indeed, CAR-T-cell therapy for OS has been explored with certain encouraging progress. Currently, the main targets of CAR-T cells for OS treatment are as follows: HER2, GD2, B7-H3, IL-11Ra, IGF1R, ROR1, NKG2D, EphA2, CD166, and CSPG4. In most of these studies, the researchers found that T cells loaded with the various CARs presented antitumor activity in vitro and in vivo preclinical models, partially associated with extended survival. Therefore, CAR-T cells may be a new therapeutic approach to treat OS. In addition, among all the targets, HER2, IL-11Ra, B7-H3, and EphA2 deserve further study because their CAR-T cells can act on metastases, which are the main causes of death in OS patients [15,29,30,86].

Although CAR-T-cell therapy is a promising approach for OS, some limits still await solutions. The problems include the following:Some targets are not tumor specific; for example, CD166 is present not only in tumors but also in some normal tissues, such as epithelial cells, fibroblasts, and neurons [115,116,117].Each target is not 100% expressed in tumor cells, and there has been antigen loss or modulation after CAR-T cell therapy [126,127]. Hsu and colleagues reported an increase in EphA2-negative tumor cells following experimental EphA2 CAR treatment of OS [30].Tumor cells may upregulate some surface receptors to induce apoptosis of CAR-T cells; Chulanetra et al. found that OS cells induced CAR-T cell apoptosis by upregulation of PD-L1 [38].Although CAR-T-cell therapy has shown encouraging results in treating leukemia and lymphoma, it is not representative of all cancer therapy; in treating most solid tumors, there are still insurmountable obstacles, including tumor trafficking and tumor microenvironment [67,128].Common adverse effects of CAR-T-cell therapy, such as cytokine storms, are also present in CAR-T-cell therapy for OS; in the Morgan RA case report, a patient died after a cytokine storm following treatment with HER2-targeted CAR-T cells [61].

To address these problems, we can adopt the following solutions:First, we can find more precise targets for tumors. Second, if a target is highly expressed in the tumor tissue and lowly expressed in the normal tissue, we can adjust the therapeutic threshold so it only acts on the tumor tissue, achieving “dose-targeting”. Third, adjusting the affinity between CAR-T cells and the tumor-associated antigen can protect the healthy tissue [129,130]. Last, a logic-gated CAR is a viable method.The problems of antigen low expression or loss and apoptosis of CAR-T cells can be solved by the following methods:Constructing immune cells expressing multiple CARs or combining multiple CAR-T cells [131].Creating combinations of CAR-T cells with conventional chemotherapeutic agents or MABs.Continuing to improve T cells, such as entering more costimulatory domains to enhance the killing ability of T cells [132] (Figure 3).To treat solid tumors, the following strategies can be adopted:Some chemokine receptors can recognize the upregulating chemokines in TME; we thus can construct CAR-T cells loaded with these chemokine receptors to increase the infiltration of CAR-T cells. In addition, we can also design CAR-T cells that disintegrate the extracellular matrix proteins forming the physical barrier to TME [133,134,135].We can construct CAR-T cells that target not only tumor antigens but also immunosuppressive cytokines or immunosuppressive cells, such as MDSCs in TME, to resist tumor immunosuppressive effects on T cells [37,136].We can strengthen CAR signaling through the proper modulation of CD3ζ immunoreceptor tyrosine-based activation motifs, which has been reported to reduce T-cell exhaustion [137].We can limit the over-potent function of T cells by introducing coexpressing suicide genes or inhibitive receptors to avoid cytokine storms [138]. However, further rigorous preclinical and clinical trials of CAR-T cells are necessary to rule out adverse effects.

Lastly, there are some novel attempts in the treatment of cancers, such as fusion genes. Chromosomal translocations have been associated with the occurrence of certain cancers. Translocation-derived fusion genes can encode some chimeric RNA, commonly transcriptional regulators, generating structurally novel oncogenic fusion proteins [139]. EWS-FLI1 chimeric transcription factor has been widely reported to be involved in the occurrence of Ewing’s sarcoma [140,141,142]. Similarly, this phenomenon has also been identified in OS [143]. The Rab22a-NeoF1 fusion gene has been shown to be expressed in OS and to promote metastasis of OS [144,145,146,147]. In addition, EWSR1-CREB3L1 has been identified as a novel fusion transcript in small-cell OS [148], and the FN1-FGFR1 fusion gene may be a target in chondroblastoma-like OS [149]. Therefore, targeting these fusion genes may be a promising strategy. Furthermore, if epitope-targeted CAR-T cells can be effectively combined with fusion gene-targeted therapy, further efficacy may be achieved.

## 4. Conclusions

CAR-T cell therapy for OS is promising, but its practicality and safety require further research.

## Figures and Tables

**Figure 1 cancers-13-04469-f001:**
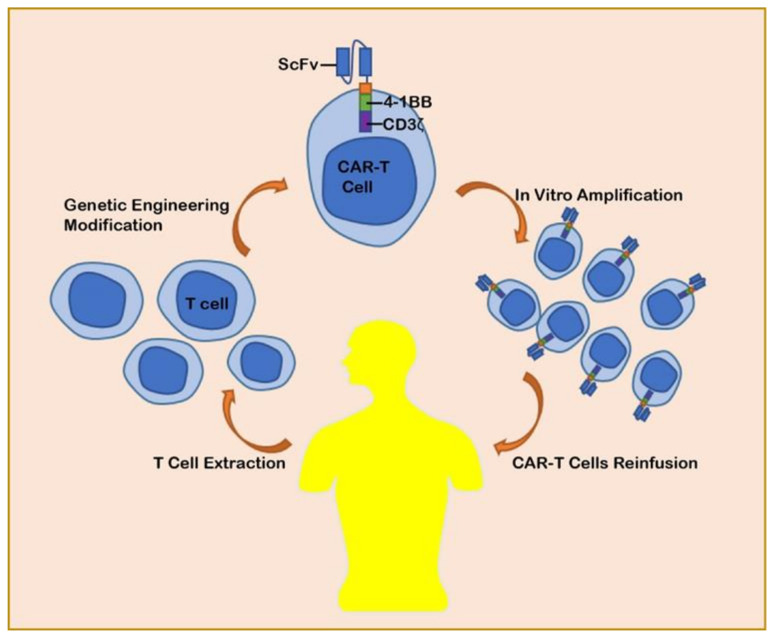
Flowchart of CAR-T cells production.

**Figure 2 cancers-13-04469-f002:**
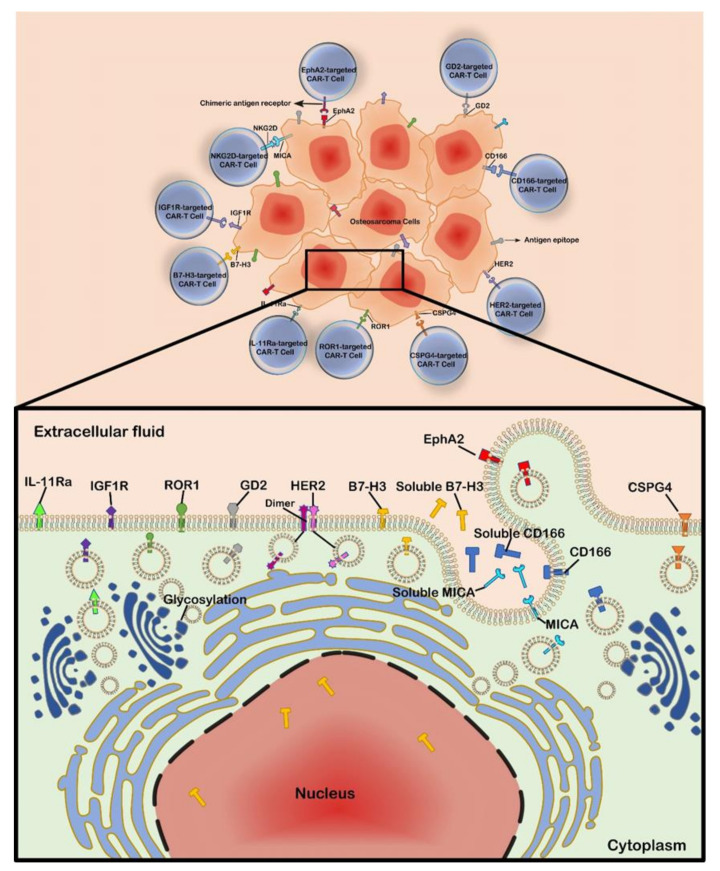
The main targets of CAR-T cells for OS treatment.

**Figure 3 cancers-13-04469-f003:**
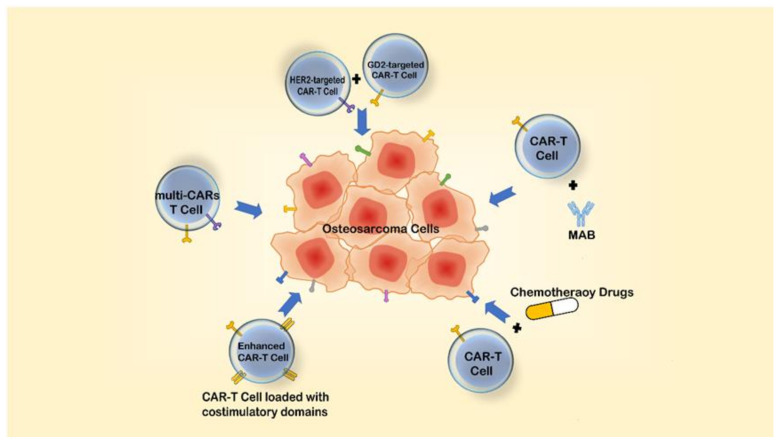
Solutions to enhance the efficacy of CAR-T cell therapy.

**Table 1 cancers-13-04469-t001:** Targets of CAR-T cells in OS.

	Characteristics	Expression in Normal Tissue	Kills OS Cells and/or Metastases	References
HER2	A tyrosine-protein kinase	Low expression in normal tissue	Kills OS cells and metastases	[15,23,31,32,33,34]
GD2	An N-acetyl neuraminic acid-containing glycolipid antigen	Low expression in normal tissue	Kills OS cells	[23,35,36,37,38]
B7-H3	A member of the B7 family of immunoregulatory proteins	Low expression in normal tissue	Kills OS cells and metastases	[29,39,40]
IL-11Ra	A member of the PI3K, MAPK and JAK-STAT activating family of cytokines/receptors	Low expression in normal tissue	Kills OS cells and metastases	[24,41]
IGF1R	A transmembrane glycoprotein	Widely distributed in normal tissues, such as myocardium, brain, bone, and cartilage	Kills OS cells	[25,42,43]
ROR1	A type I transmembrane protein	Expression in normal tissue, particularly high in the gastric antrum and body	Kills OS cells	[25,44,45]
NKG2D	A powerful activating receptor expressed by natural killer (NK) cells and T cells	Expressed by NK cells and T cells	Kills OS cells	[46,47]
EphA2	A tyrosine kinase receptor	Mainly confined to some epithelial cells	Kills OS cells and metastases	[30,48,49,50]
CD166	A type I membrane protein, a member of the immunoglobulin gene superfamily, and a ligand for the lymphocyte antigen CD6	Broadly expressed in various tissues and cells, including neuronal, immune, and epithelial cells, as well as stem cells of hematopoietic and mesenchymal origin	Kills OS cells	[27,51,52]
CSPG4	A transmembrane proteoglycan	Low expression in normal tissue	Kills OS cells	[53,54,55]

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
