# Peer review of "Chimeric Antigen Receptor T-Cell Therapy: The Light of Day for Osteosarcoma"

_cancers, 2021, doi:10.3390/cancers13174469_

Round 1
Reviewer 1 Report
This review is interesting and has potential value for osteosarcoma. But I wonder if have some novel antigens for immunity therapy. and need to add some novel oncogene or fusion gene report and some other breakthrough progress. I suggest the authors consider to focus on the chimeric RNA and oncogene on sarcoma
Author Response
Thank you for your professional suggestions. Recently, more and more studies have begun to focus on fusion genes in the occurrence and development of cancers, and fusion genes also show a strong therapeutic prospect in the treatment of cancers. Your comments are valuable and very helpful for revising and improving our paper. Therefore, in the revised manuscript, we added fusion genes in the discussion section according to your suggestions.
Reviewer 2 Report
Although the CAR-T cells therapy is highly effective for blood cancers, its efficacy in solid tumor (such as osteosarcoma) treatment has not yet been supported; no single, superior target molecule, such as CD19 in blood cancers, has not been found in solid tumors. This article has reviewed and clearly summarized the safety, specificity and clinical transformation of the targets in treating osteosarcoma.
Author Response
Thank you for your professional and valuable comments. Indeed, the CAR-T cells therapy is highly effective for blood cancers, while, its efficacy in solid tumour (such as osteosarcoma) treatment has not yet been supported. One important reason is that there is not much research on this aspect, which is only in the exploratory stage. Therefore, we reviewed the latest literature about the CAR-T cells therapy in osteosarcoma, and summarized the safety, specificity and clinical transformation of the targets in treating, and points out the direction of further exploration. We hope that this article will provide useful information about CAR-T cells therapy in the treatment of osteosarcoma to experts in relevant fields.
Reviewer 3 Report
Dear Editor, dear Authors,
The manuscript I have been called to review describes the state of the art of the use of CAR therapy in the treatment of Osteosarcoma. Overall, this review offers good starting points for the development of a new generation of CARs able to overcome the off-target effects observed in clinical trials.
I have just two major concerns regarding this manuscript.
- The first one is related to the, sometimes not really proper, English style used by the authors. I made some corrections, but I am not an English mother tongue, so I think it could be better to ask the contribution of a mother tongue to help the authors to revise the manuscript.
- I think that a Figure representing the different constituents of the CARs, and how CARs are engineered might help the reader to better understand what the authors talk about.
Minor:
31: It seems that a verb is missing
32-34: Rephrase this sentence
35: composed by, instead of "that compose"
38: Please, give meaning to the acronymous CAR-T. Only CAR has been explained.
37-42: To me, it is not very clear, what CAR-T is. Please, simplify this sentence and make it more focused and clear. For example, the authors do not tell that CAR is placed in the membrane of T cells (this is written later), but they describe it as the reader already knows it.
48: Remove "And" after the point
113: substitute "leading" with "... which are responsible for the death of..." or something similar.
133: Delete " in experiment by" and leave "Adrienne H. Long et al. observed 100% ..."
147: xenografts from pediatric sarcoma
234: Delete "And" at the beginning of the sentence
238: Please check the sentence "Besides, their studies have shown ROR1-targeted CAR-T cells showed specific cytotoxicity..."
371: "Addressing these problems, we can adopt the following solutions can", please rephrase.
381: Figure 2 is referred to the previous sentence. Please remove the point after "[132].".
381: "To treat solid tumors, the following strategies can be adopted.". Please substitute the point at the end of the sentence with a colon mark.
371-393: please revise the numbered list, sometimes numbers are followed by brackets, other by points. Please always use capital characters at the beginning of the sentences.
Author Response
Point 1: The first one is related to the, sometimes not really proper, English style used by the authors. I made some corrections, but I am not an English mother tongue, so I think it could be better to ask the contribution of a mother tongue to help the authors to revise the manuscript.
Response 1: Thank you for your professional suggestion. Our revised version has been edited and modified by professional language editing agencies. Inappropriate language expression is a deficiency of this manuscript. We are also trying our best to modify the language to meet the publishing requirements. After in-depth language modification, the language of this manuscript has been more in line with the expression habits of English mother tongue. I hope our efforts can be recognized by you. Thank you again for your valuable suggestions.
Point 2: I think that a Figure representing the different constituents of the CARs, and how CARs are engineered might help the reader to better understand what the authors talk about.
Response 2: Thank you for your professional suggestion. We designed a picture to illustrate the production process of CAR-T cells and briefly described it in the revised version.
Response 3: Thank you for your professional and valuable comments. We have made corresponding changes in the revised vision according to your comments. Besides, we use MDPI editing service for the revised vision.
31: It seems that a verb is missing.
Done.
32-34: Rephrase this sentence
Done.
35: composed by, instead of "that compose"
Done.
38: Please, give meaning to the acronymous CAR-T. Only CAR has been explained.
Done.
37-42: To me, it is not very clear, what CAR-T is. Please, simplify this sentence and make it more focused and clear. For example, the authors do not tell that CAR is placed in the membrane of T cells (this is written later), but they describe it as the reader already knows it.
Thank you for your professional suggestion. We designed a picture to illustrate the production process of CAR-T cells and briefly described it in the revised version.
48: Remove "And" after the point
Done.
113: substitute "leading" with "... which are responsible for the death of..." or something similar.
Done.
133: Delete " in experiment by" and leave "Adrienne H. Long et al. observed 100% ..."
Done.
147: xenografts from pediatric sarcoma
Done.
234: Delete "And" at the beginning of the sentence
Done.
238: Please check the sentence "Besides, their studies have shown ROR1-targeted CAR-T cells showed specific cytotoxicity..."
Thank you for your professional suggestions. Xin Huang et al did find that ROR1-targeted CAR-T cells exhibited a specific cytotoxicity to sarcoma cells.
371: "Addressing these problems, we can adopt the following solutions can", please rephrase.
Done.
381: Figure 2 is referred to the previous sentence. Please remove the point after "[132].".
Done.
381: "To treat solid tumors, the following strategies can be adopted.". Please substitute the point at the end of the sentence with a colon mark.
Done.
371-393: please revise the numbered list, sometimes numbers are followed by brackets, other by points. Please always use capital characters at the beginning of the sentences.
Done.
